# Psychosocial stress and neuroendocrine biomarker concentrations among women living with or without HIV

**Matthew E. Levy**[1,2]*, **Ansley Waters**[1,3], **Sabyasachi Sen**[4], **Amanda D. Castel**[1], **Michael Plankey**[5], **Sherry Molock**[6], **Federico Asch**[7], **Lakshmi Goparaju**[5], **Seble Kassaye**[5]

**1** Department of Epidemiology, Milken Institute School of Public Health at the George Washington University, Washington, DC, United States of America, **2** Westat, Rockville, Maryland, United States of America, **3** Division of Clinical Epidemiology, Office of Epidemiology, Virginia Department of Health, Richmond, Virginia, United States of America, **4** Division of Endocrinology, George Washington University School of Medicine and Health Sciences, Washington, DC, United States of America, **5** Department of Medicine, Georgetown University Medical Center, Washington, DC, United States of America, **6** Department of Psychology, The George Washington University, Washington, DC, United States of America, **7** Cardiovascular Core Laboratories and Cardiac Imaging Research, MedStar Health Research Institute, MedStar Heart and Vascular Institute, Washington, DC, United States of America

* mattelevy@gwu.edu

## Abstract

### Objective

Women living with HIV (WLWH) experience psychosocial stress related to social-structural vulnerabilities. To investigate neuroendocrine pathways linking stress and increased cardio-vascular disease risk among WLWH, we evaluated associations between psychosocial stress (i.e., perceived stress, posttraumatic stress, and experiences of race- and gender-based harassment) and a composite neuroendocrine biomarker index among WLWH and women without HIV.

### Methods

In 2019–2020, Women's Interagency HIV Study participants in Washington, DC completed a questionnaire and provided blood and 12-hour overnight urine samples for testing of serum dehydroepiandrosterone sulfate (DHEA-S) and urinary free cortisol, epinephrine, and norepinephrine. Psychosocial stress was measured using the Perceived Stress Scale, PTSD Checklist-Civilian Version, and Racialized Sexual Harassment Scale. Latent profile analysis was used to classify participants into low (38%), moderate (44%), and high (18%) stress groups. Composite biomarker index scores between 0–4 were assigned based on participants' number of neuroendocrine biomarkers in high-risk quartiles ($\geq 75$th percentile for cortisol, epinephrine, and norepinephrine and $\leq 25$th percentile for DHEA-S). We evaluated associations between latent profile and composite biomarker index values using multivariable linear regression, adjusting for socio-demographic, behavioral, metabolic, and HIV-related factors.

**Data Availability Statement:** Data are from the Women's Interagency HIV Study (WIHS), now the MACS/WIHS Combined Cohort Study (MWCCS). These data cannot be freely and publicly shared as

they are of a highly sensitive nature. Data are available to investigators in two ways. First, the MWCCS Public Use Data Set is available upon request. This data set provides de-identified data (meeting HIPAA criteria) that may assist anyone interested in public health research. Access to the MWCCS Public Data Set may be obtained by filling out the MWCCS Public Data Set Request form at https://statepi.jhsph.edu/wihs/wordpress/?page_id=10771/. Alternatively, the MWCCS welcomes collaborations with investigators and with other cohorts, both nationally and internationally, who can access the entire richness of data and specimens that are available. To collaborate, a concept sheet must be submitted, reviewed, and approved by the MWCCS Executive Committee. Details on how to submit a concept sheet can be found at https://statepi.jhsph.edu/mwccs/work-with-us/. This is a requirement of cohort IRB approvals ensuring secure, timely, and ethical sharing of the cohort's data. Questions about how to access data may be directed to the MWCCS Data Analysis and Coordination Center (mwccs@jhu.edu) or to senior author Dr. Seble Kassaye (sgk23@georgetown.edu).

**Funding:** This research was funded in part by a 2019 pilot award from the District of Columbia Center for AIDS Research (DC CFAR), a National Institutes of Health (NIH)-funded program (P30AI117970), which is supported by the following NIH Co-Funding and Participating Institutes and Centers: National Institute of Allergy and Infectious Diseases (NIAID); National Cancer Institute (NCI); Eunice Kennedy Shriver National Institute of Child Health and Human Development (NICHD); National Heart, Lung, and Blood Institute (NHLBI); National Institute on Drug Abuse (NIDA); National Institute of Mental Health (NIMH); National Institute on Aging (NIA); National Institute of Diabetes and Digestive and Kidney Diseases (NIDDK); National Institute on Minority Health and Health Disparities (NIMHD); National Institute of Dental and Craniofacial Research (NIDCR); National Institute of Nursing Research (NINR); Fogarty International Center (FIC); and Office of AIDS Research (OAR). The contents of this publication are solely the responsibility of the authors and do not represent the official views of the NIH. Data in this manuscript were collected by the Women's Interagency HIV Study (WIHS), now the MACS/WIHS Combined Cohort Study (MWCCS). MWCCS (Principal Investigators): Data Analysis and Coordination Center (Gypsyamber D'Souza, Stephen Gange, and Elizabeth Golub), U01-HL146193; and Metropolitan Washington Clinical Research Site (Seble Kassaye and Daniel Merenstein), U01-HL146205. The MWCCS is

## Results

Among 90 women, 62% were WLWH, 53% were non-Hispanic Black, and median age was 55 years. In full multivariable models, there was no statistically significant association between psychosocial stress and composite biomarker index values among all women independent of HIV status. High (vs. low) psychosocial stress was positively associated with higher mean composite biomarker index values among all monoracial Black women (adjusted β = 1.32; 95% CI: 0.20–2.43), Black WLWH (adjusted β = 1.93; 95% CI: 0.02–3.83) and Black HIV-negative women (adjusted β = 2.54; 95% CI: 0.41–4.67).

## Conclusions

Despite a null association in the overall sample, greater psychosocial stress was positively associated with higher neuroendocrine biomarker concentrations among Black women, highlighting a plausible mechanism by which psychosocial stress could contribute to cardiovascular disease risk.

## Introduction

Over the last two decades, the aging of people living with HIV (PLWH) and the widespread use of antiretroviral therapy have been accompanied by a decrease in AIDS-related deaths and a concomitant increase in deaths attributed to cardiovascular disease (CVD) and other non-AIDS-related chronic conditions [1, 2]. PLWH have an elevated risk of subclinical atherosclerosis and atherosclerotic CVD, attributed in large part to chronic inflammation and immune activation [3–5]. Cisgender women living with HIV (WLWH) have an approximately three-times greater CVD risk compared with HIV-negative women [6–9], suggesting that the HIV-associated CVD risk is greater among women compared with men, the etiology of which is not well understood [10].

Psychosocial stress is one CVD risk pathway that remains understudied among WLWH, despite WLWH and particularly WLWH of color being disproportionately affected [11]. Forms of psychosocial stress experienced by WLWH include those directly related to managing one's HIV diagnosis (e.g., rigid medication regimens, side effects, sense of physical vulnerability based on immunodeficiency), those indirectly related (e.g., stigma, emotional challenges), and intersecting social-structural vulnerabilities related to interdependent and mutually constitutive identities such as being female, Black or Hispanic, and HIV-positive [12–19]. Experiences of abuse and trauma are also common among WLWH. In a meta-analysis of studies conducted among WLWH, there was a high estimated overall prevalence of intimate partner violence (55%), sexual abuse (35%), physical abuse (55%), and recent posttraumatic stress disorder (PTSD) (30%) [11].

At the end of 2019, 57% of all WLWH in the United States were Black women [20]. Among Black women, discrimination attributed to racism and sexism is common and is associated with elevated stress [21, 22]. Gendered racism, or the simultaneous experience of both racism and sexism, is an inherent stressor that can negatively affect personal relationships, professional and economic opportunities, and mental and physical health outcomes among Black women [23–26]. Gendered racism causes psychosocial stress both directly and through increased vulnerability to other stressors [21, 23–25]. Further, among Black WLWH, the

funded primarily by NHLBI, with additional co-funding from NICHD, NIA, NIDCR, NIAID, National Institute of Neurological Disorders and Stroke (NINDS), NIMH, NIDA, NINR, NCI, National Institute on Alcohol Abuse and Alcoholism (NIAAA), National Institute on Deafness and Other Communication Disorders (NIDCD), NIDDK, NIMHD, and in coordination and alignment with the research priorities of the NIH OAR. The funders provided support in the form of salaries for authors [ML, AW, MP, LG, SK], but did not have any additional role in the study design, data collection and analysis, decision to publish, or preparation of the manuscript. The specific roles of these authors are articulated in the 'author contributions' section. One of the authors [ML] is currently affiliated with a commercial entity, Westat. Data collection and analysis for this study occurred prior to his current affiliation with Westat, while he was affiliated with The George Washington University.

**Competing interests:** One of the authors [ML] is currently affiliated with a commercial entity, Westat. Data collection and analysis for this study occurred prior to his current affiliation with Westat, while he was affiliated with The George Washington University. This does not alter our adherence to PLOS ONE policies on sharing data and materials.

effects of gendered racism and HIV-based discrimination can compound one another, representing key social-structural barriers to HIV care [27].

Psychosocial stress is known to contribute to atherosclerosis through heightened activity of the hypothalamic pituitary adrenal (HPA) axis and sympathetic nervous system (SNS) [28]. This physiologic response to perceived stress involves the release of glucocorticoid hormones such as cortisol and the production of epinephrine and norepinephrine, which collectively raise blood pressure, reduce insulin sensitivity, decrease hemostasis, and cause endothelial dysfunction [29]. Psychosocial stress is also associated with adverse health behaviors including physical inactivity, suboptimal diet, and smoking [30–33]. Among WLWH, higher levels of psychosocial stress were positively associated with subclinical carotid atherosclerosis [34]. In the general population, chronic daily stressors such as work-related stress as well as PTSD symptoms have been associated with increased CVD risk [35, 36]. Racial discrimination as a chronic stressor has also been identified as a key contributor of increased CVD risk among Black individuals [37–40].

Largely outside the context of HIV-related research, neuroendocrine biomarkers have been measured as part of a larger set of biomarkers collectively used to measure allostatic load, or the dysregulation of multiple biologic systems, including the neuroendocrine system, as a result of wear and tear brought on by cumulative exposure to stress [41–47]. The four neuroendocrine biomarkers most commonly evaluated have included cortisol and dehydroepiandrosterone sulfate (DHEA-S) as markers of HPA axis activity and epinephrine and norepinephrine as markers of SNS activity [48]. Higher levels of these biomarkers (lower for DHEA-S) under basal (resting) states have been taken to indicate a chronic heightened neuroendocrine response that might be attributed to psychosocial stress. Among women in one study, current life stressors were associated with more extreme levels of these four biomarkers [49]. These biomarkers have also been associated with increased CVD risk [50–52].

Several studies conducted among PLWH have employed a composite biomarker index for allostatic load that incorporated the aforementioned neuroendocrine biomarkers along with measures for inflammatory (e.g., C-reactive protein), metabolic (e.g., glucose), and cardiovascular (e.g., heart rate) markers [53–57]. To our knowledge, there have been no published studies specifically examining a composite neuroendocrine biomarker index and its correlates among WLWH. The objective of this study was to evaluate associations between psychosocial stress (i.e., perceived stress, posttraumatic stress, and experiences of race- and gender-based harassment) and a composite neuroendocrine biomarker index among WLWH and HIV-negative women. We focused on neuroendocrine biomarkers because they are proximal markers of the physiologic stress response and serve as mediators of the effects of stress on inflammation and metabolic and cardiovascular dysfunction [28, 58, 59]. We hypothesized that this association would be stronger among WLWH compared with HIV-negative women. Given that Black WLWH disproportionately experience stress related to intersecting social-structural vulnerabilities [60–62], we also conducted a secondary subgroup analysis in which we evaluated this association specifically among Black women. Psychosocial correlates of neuroendocrine biomarkers could provide potential targets for interventions to prevent CVD among WLWH.

## Methods

### Study population

This cross-sectional sub-study was conducted within the Women's Interagency HIV Study (WIHS) at the Washington, DC site. The WIHS is an ongoing multicenter prospective observational study of WLWH and demographically similar women without HIV who were eligible

based on sexually transmitted infection history, intravenous drug use, and/or sexual risk behaviors [63–65]. Participants were initially recruited and enrolled in the WIHS in 1994–1995, 2001–2002, or 2011–2013. At semi-annual visits, data were collected through structured interviews and standardized physical, psychological, and laboratory assessments. HIV status was assessed by enzyme-linked immunosorbent assay (ELISA) with Western blot for confirmation at study enrollment for WLWH, and semi-annually for HIV-negative participants. The WIHS study protocol was approved by the Georgetown University Institutional Review Board (IRB). All participants provided written informed consent.

Between August 2019 and January 2020, 90 participants were enrolled in this sub-study, including 56 WLWH (62% of the sample) and 34 HIV-negative women (38% of the sample). Participants were recruited at their routine semi-annual visits or by phone. We administered a brief screening questionnaire to evaluate potential participants' eligibility and their ability to complete study procedures. Women were eligible if they reported no non-topical use of hormone supplements or therapy in the last three months (e.g., corticosteroids, epinephrine, norepinephrine, dehydroepiandrosterone, estrogen, progesterone, and their derivatives). For participants screened by phone, this eligibility criterion was re-confirmed at their study visit. The protocol for this sub-study was approved by Georgetown University and George Washington University IRBs. All participants provided written informed consent.

## Study procedures

**Visit procedures.**   At the study visit, participants completed a questionnaire on perceived stress, posttraumatic stress, and experiences of race- and gender-based harassment (see *Psychosocial Stress*), provided a blood sample, and received materials for home-based collection of a 12-hour overnight urine sample. The 15-minute questionnaire was either self-administered on a tablet computer or administered by a trained interviewer, per participants' preference. One 3-mL serum-separating tube of blood was collected from each participant. Women scheduled their overnight urine collection for a night of their choice, preferably within one week of their visit and within a maximum of 30 days. Supplies provided included a 3-L urine collection container with 25 mL of 6N hydrochloric acid preservative, a toilet 'hat,' an insulated cooler bag, and an ice pack. Participants were also provided with detailed verbal and written instructions.

**Home-based urine collection.**   We contacted participants by phone with reminders of their scheduled urine collection and asked them to reschedule if they were menstruating or feeling ill. We also asked participants to refrain from strenuous activity, sexual intercourse, alcohol use, and non-prescribed substance use during the collection period. On a urine collection log, participants recorded their collection start and end times and all void times during the 12-hour period. Participants were asked to void urine at 7 pm and to collect all subsequent urine until 7 am the following day. A 12-hour collection period was chosen over a 24-hour collection period to maximize feasibility and compliance and to obtain biomarker measurements under basal (resting) conditions over a relatively non-stimulated period. Participants could store their sample in their refrigerator or in the cooler bag provided. On the day on which their collection ended, participants transported their sample to the clinic using the cooler bag and ice pack. Following specimen receipt, we administered a second questionnaire to assess compliance with the urine collection protocol and collect data on behaviors and other factors that could have influenced urinary concentrations of free cortisol or catecholamines (e.g., substance use, illness). Participants received $75 for completing the study.

**Laboratory procedures.**   Following blood collection, the serum-separating tube was inverted five times, allowed to clot upright for 30 minutes, and centrifuged at 1100–1300 x g

for 10–15 minutes. For urine samples, we measured the volume, inverted the container five times, and aliquoted the sample into two 10-mL vials for testing of free cortisol and catecholamines. The blood sample was shipped at room temperature on the collection day and an immunoassay was performed to determine the serum concentration of DHEA-S (in µg/dL). The urine vials were shipped in a refrigerated shipping container on the same day as specimen receipt. Concentration of urinary free cortisol (in µg/12 hrs) was determined using liquid chromatography/tandem mass spectrometry and concentrations of urinary fractionated catecholamines (i.e., epinephrine and norepinephrine) (in µg/12 hrs) were determined using high performance liquid chromatography. All laboratory testing was performed at Quest Diagnostics (Baltimore, MD) using standard clinical laboratory methods. Urinary concentrations were standardized by grams of creatinine because dissimilar body size leads to differential concentrations in the urine.

## Composite neuroendocrine biomarker index

We created a composite neuroendocrine biomarker index using laboratory test results for the four biomarkers of interest. Since there are no accepted thresholds for defining 'high-risk' levels that signify a heightened neuroendocrine response, the high-risk range has commonly been defined using distribution percentiles–often, the 25th and 75th percentiles [41–47]. Following this convention, we assigned one 'point' for each test result in the high-risk quartile, that is, the upper quartile ($\geq$75th percentile) for urinary free cortisol and catecholamines and the lower quartile ($\leq$25th percentile) for serum DHEA-S. Thus, values of the biomarker index could range from 0–4, representing the number of biomarkers for each participant that were in the high-risk quartile. Although we could not establish whether biomarker concentrations were abnormal using percentile cutoffs, this approach allowed us to evaluate whether higher levels of psychosocial stress were associated with more extreme biomarker concentrations within the sample. However, because individuals with PTSD have been found to have low resting cortisol, we also conducted a sensitivity analysis in which we used a bidirectional cutoff approach for defining high-risk levels for urinary cortisol (i.e., $\leq$12.5th or $\geq$87.5th percentiles) [66, 67].

## Psychosocial stress

Three different types of psychosocial stress were measured via self-report: perceived stress, posttraumatic stress, and experiences of race- and gender-based harassment. Using latent profile analysis [68], we created three mutually exclusive groups (i.e., low, moderate, and high levels of psychosocial stress) based on participants' reported levels of each type of psychosocial stress (see *Statistical Analysis*).

**Perceived stress.**   Perceived stress was measured using the Perceived Stress Scale (PSS-10), which assesses the degree to which life situations in the previous month are appraised as unpredictable, uncontrollable, and overloaded [69]. Ten items are rated on a 5-point Likert scale from "never" (assigned a score of 0) to "very often" (assigned a score of 4), with four positively stated items reverse-coded ($\alpha = 0.86$). Items assessing the frequency of perceived stress include "Felt that you were unable to control the important things in your life" and "Found that you could not cope with all the things that you had to do." The summarized score has a possible range of 0–40.

**Posttraumatic stress.**   Posttraumatic stress was measured using the PTSD Checklist-Civilian Version (PCL-C), which assesses PTSD symptoms in the last month, including re-experiencing, avoidance, and hyperarousal, as defined by the *Diagnostic and Statistical Manual of Mental Disorders, Fourth Edition* (DSM-IV) [70]. The PCL-C asks about symptoms in relation to generic stressful experiences rather than in relation to a specific index event. Although

this aspect of the PCL-C facilitates its use in different populations, it also limits its ability to identify sources of stress and determine whether stressful experiences meet criteria for a traumatic event. Seventeen items are rated on a 5-point Likert scale from "not at all" (assigned a score of 1) to "extremely" (assigned a score of 5) ($\alpha = 0.95$). Items assessing the frequency of symptoms include "Repeated, disturbing memories, thoughts, or images of a stressful experience," "Avoiding activities or situations because they reminded you of a stressful experience," and "Feeling distant or cut off from other people." The summarized score has a possible range of 17–85.

**Experiences of race- and gender-based harassment.** Experiences of race- and gender-based harassment in the last year were measured using the Racialized Sexual Harassment Scale (RSHS), which assesses harassment related to one's race or ethnicity, harassment related to one's gender, and harassment related simultaneously to one's race or ethnicity and gender [71]. Twenty-one items are rated on a 5-point Likert scale from "never" (assigned a score of 0) to "very often" (assigned a score of 4) ($\alpha = 0.97$). Items assessing the frequency of experiences include "Someone said things to insult people of your race/ethnicity (for example, saying people of your race/ethnicity can't handle certain jobs)," "Someone made comments about your body that emphasized your gender (for example, comments about the size of your breasts or penis)," and "Someone said they expected you to behave certain ways because of your gender and ethnicity (for example, expected you as a Black or Latina woman to wear inappropriate clothes, expected you as an Asian man to be self-controlled and disciplined, as an Asian woman to try to please others, as a Latino man that you would be unfaithful in relationships, etc.)." The summarized score has a possible range of 0–84.

## Covariates

Covariates were measured at participants' most recent core study visit prior to the visit conducted for this sub-study. The median duration between participants' core and sub-study visits was 162 days (interquartile range [IQR]: 125–194; range: 0–275). Socio-demographic and behavioral factors were self-reported and included age, race/ethnicity, education upon study entry (did not complete high school, completed high school, or attended/completed college), income ($\leq$\$30,000 or >\$30,000 per year), current smoking, current alcohol use (abstainer, 1–7 drinks/week, 8–12 drinks/week, or $\geq$13 drinks/week), recent substance use since last visit (crack/cocaine, heroin, or methamphetamines), and current use of a prescribed psychotropic medication. Postmenopausal status was self-reported and defined as not having menstruated in 12 or more months, unless related to pregnancy or medication use. Body mass index (BMI) in $kg/m^2$ was calculated based on weight and height measurements. HIV-related measures included current use of antiretroviral therapy, current CD4+ T cell count, and current HIV RNA viral load (<20 copies/mL or $\geq$20 copies/mL). If covariates had missing values at the most recent visit, then the most recent non-missing value was carried forward from a prior visit (for a total of n = 1 for income; n = 1 for smoking, alcohol use, and substance use; n = 11 for psychotropic medication use; n = 1 for menopausal status; n = 1 for CD4+ cell count; and n = 3 for HIV viral load).

## Statistical analysis

Socio-demographics, behavioral factors, psychosocial stress measures, and neuroendocrine biomarker concentrations were summarized using frequencies and proportions or medians and IQRs and were compared by HIV serostatus using Chi-square, Fisher's exact, or Wilcoxon rank sum tests. Pairwise correlations were evaluated among the three psychosocial stress measures using Spearman's correlation coefficients. We assessed whether differences in

**Table 1. Fit statistics for latent profile models of psychosocial stress among women in the Women's Interagency HIV Study, Washington, DC, 2019–2020 (n = 90).**

| Fit statistics | 2 classes | 3 classes | 4 classes |
|---|---|---|---|
| AIC | 625.9 | **605.1** | 611.5 |
| BIC | 650.9 | **640.1** | 656.5 |
| ABIC | 619.3 | **595.9** | 599.7 |
| Adj. LMR-LRT (*p*) | **68.7 (0.0002)** | **27.3 (0.0049)** | 1.5 (0.72) |
| BLRT (*p*) | **72.6 (<0.0001)** | **28.8 (<0.0001)** | 1.5 (1.0) |
| Entropy | 0.807 | **0.886** | 0.790 |

Abbreviations: ABIC, sample size adjusted Bayesian information criterion; AIC, Akaike information criterion; BIC, Bayesian information criterion; BLRT, bootstrap likelihood ratio test; adj. LMR-LRT, adjusted Lo-Mendell-Rubin likelihood ratio test; *p*, p-value.

Boldface indicates optimal fit (i.e., lowest for AIC, BIC, and ABIC; p<0.05 for adj. LMR-LRT and BLRT; highest for entropy).

neuroendocrine biomarker concentrations between WLWH and women without HIV persisted after adjustment for confounders using multivariable linear regression. Psychosocial stress measures and neuroendocrine biomarker concentrations were also compared among racial/ethnic subgroups using Fisher's exact or Kruskal-Wallis tests.

We used latent profile analysis to classify participants into mutually exclusive psychosocial stress groups based on participants' PSS-10, PCL-C, and RSHS scores. Latent profile analysis is a classification method that identifies discrete profiles (or classes) of individuals who share similar response patterns to a set of continuous variables [68]. Using MPlus Version 7.4, we considered models with 2–4 classes. The best fitting model was determined by several criteria: lower Akaike information criterion (AIC), Bayesian information criterion (BIC), and sample-size adjusted BIC values; statistically significant p-values for the adjusted Lo-Mendell-Rubin and boot-strapped likelihood ratio tests; and higher entropy values. A three-class model was preferable over two- and four-class models (Table 1). We assigned participants to the latent profile for which they had the highest posterior probability of membership. We named the profiles 'low psychosocial stress' (38% of participants assigned), 'moderate psychosocial stress' (44% of participants assigned), and 'high psychosocial stress' (18% of participants assigned), based on the distribution of PSS-10, PCL-C, and RSHS scores in each class (Table 2). The

**Table 2. Patterns of perceived stress, posttraumatic stress, and race- and gender-based harassment by latent profile membership (3-class model).**

|  | Latent profile 1: low psychosocial stress (n = 34) | Latent profile 2: moderate psychosocial stress (n = 40) | Latent profile 3: high psychosocial stress (n = 16) |
|---|---|---|---|
|  | median (IQR) | median (IQR) | median (IQR) |
| PSS-10[a] | 11.5 (8–16) | 17 (14.5–21) | 26 (22.5–29.5) |
| PCL-C[b] | 24 (21–27) | 44.5 (40.5–51.5) | 67.5 (64–71) |
| RSHS[c] | 6.5 (0–21) | 21 (6–34) | 34.5 (22.5–52) |

Abbreviations: IQR, interquartile range; PSS-10, PCL-C, PTSD Checklist-Civilian Version; Perceived Stress Scale; RSHS, Racialized Sexual Harassment Scale.

[a] The PSS-10 is a 10-item, 5-point Likert scale that measures perceived stress in the last month. Response options range from "never" (assigned a score of 0) to "very often" (assigned a score of 4) and four positively stated items are reverse-coded. The possible range of the total score is 0–40.

[b] The PCL-C is a 17-item, 5-point Likert scale that measures symptoms of posttraumatic stress disorder in the last month. Response options range from "not at all" (assigned a score of 1) to "extremely" (assigned a score of 5). The possible range of the total score is 17–85.

[c] The RSHS is a 21-item, 5-point Likert scale that measures experiences of race- and gender-based harassment in the last year. Response options range from "never" (assigned a score of 0) to "very often" (assigned a score of 4). The possible range of the total score is 0–84.

mean probability of membership for each respective profile was 0.98 for women in the low psychosocial stress profile, 0.91 for women in the moderate psychosocial stress profile, and 0.99 for women in the high psychosocial stress profile.

We used linear regression, overall and stratified by HIV serostatus, to evaluate the association between latent class and the composite biomarker index. Although ordinal regression is a possible alternative, because the difference between the scores of 0 and 1, 1 and 2, 2 and 3, and 3 and 4 have the same interpretation (i.e., one additional biomarker in the high-risk range), we were able to appropriately harness the greater statistical power of linear regression for this analysis. This increased power is particularly useful in this study with a smaller sample size. This approach has been applied successfully elsewhere [72–76]. In full multivariable models, we adjusted for socio-demographics, behavioral factors, menopausal status, BMI, and CD4+ cell count and HIV viral load (among WLWH only). Given the sample size, we also created parsimonious models by removing covariates with the highest p-values from full multivariable models in a stepwise manner until the maximum adjusted R-squared value was reached, beyond which further covariate removal would result in lower adjusted R-squared values. Age, race/ethnicity, and HIV status (in the overall sample) were retained in parsimonious models regardless of p-values. We also assessed associations between CD4+ cell count/HIV viral load and the composite biomarker index.

Regression analyses were further conducted separately among all non-Hispanic Black participants, non-Hispanic Black WLWH, and non-Hispanic Black HIV-negative women. Recognizing that Black individuals who identify with at least one other race (i.e., biracial, mixed-race, or multiracial individuals) are often racialized as Black, yet have unique experiences associated with existing at the intersections of multiple racial identities, we conducted these subgroup analyses once among participants who identified exclusively as Black and again among all participants who identified as Black regardless of whether they also identified with another race. There were insufficient sample sizes to conduct similar analyses among other racial/ethnic subgroups.

In sensitivity analyses, we repeated analyses with data excluded for participants who reported either not collecting all urine during the 12-hour period or use of substances on the day of urine collection that could have influenced biomarker concentrations (i.e., heavy alcohol use, crack cocaine use). P-values <0.05 were considered statistically significant. Statistical analyses (except latent profile analysis) were conducted using SAS Version 9.4.

## Results

### Participant characteristics

Among 90 participants, 56 (62%) were WLWH. Characteristics are provided in Table 3 for the full sample and stratified by HIV serostatus. Socio-demographics were similar between WLWH and HIV-negative women. Overall, the median age was 55 years (IQR: 49–62; range: 36–73), 53% were non-Hispanic Black, 34% were multiracial, 6% were non-Hispanic white, and 6% were Hispanic. One-third of participants were current smokers, 8% reported recent substance use, and 27% reported current use of a prescribed psychotropic medication. WLWH were more likely than HIV-negative women to abstain from alcohol use (66% vs. 27%). Nearly three-quarters of women (72%) were postmenopausal and median BMI was 32.6 kg/m$^2$ (IQR: 28.5–38.4). Among WLWH, 91% currently used antiretroviral therapy (66% with an integrase strand transfer inhibitor; 30% with a non-nucleoside reverse transcriptase inhibitor; 16% with a protease inhibitor), 86% had CD4+ cell counts >500 cells/μL, and 89% had HIV viral loads <20 copies/mL.

**Table 3. Characteristics by HIV serostatus among women in the Women's Interagency HIV Study, Washington, DC, 2019–2020 (n = 90).**

|  | Overall (n = 90) | WLWH (n = 56) | HIV-negative women (n = 34) |  |
|---|---|---|---|---|
|  | n (%) | n (col %) | n (col %) | p |
| **Socio-demographics** |  |  |  |  |
| Age (years), median (IQR) | 55 (49–62) | 54.5 (50–62) | 55.5 (49–61) | 0.69 |
| Race/ethnicity |  |  |  | 0.79[a] |
| Non-Hispanic Black | 48 (53.3) | 31 (55.4) | 17 (50.0) |  |
| Non-Hispanic white | 5 (5.6) | 4 (7.1) | 1 (2.9) |  |
| Hispanic | 5 (5.6) | 3 (5.6) | 2 (5.9) |  |
| Other[b] | 32 (35.6) | 18 (32.1) | 14 (41.2) |  |
| Education at study entry |  |  |  | 0.56 |
| Did not complete high school | 14 (15.6) | 10 (17.9) | 4 (11.8) |  |
| Completed high school | 37 (41.1) | 24 (42.9) | 13 (38.2) |  |
| Attended or completed college | 39 (53.3) | 22 (39.3) | 17 (50.0) |  |
| Income ≤$30,000 per year | 62 (68.9) | 41 (73.2) | 21 (61.8) | 0.26 |
| **Behavioral characteristics** |  |  |  |  |
| Current smoker | 30 (33.3) | 17 (30.4) | 13 (38.2) | 0.44 |
| Current alcohol use |  |  |  | 0.0004[a] |
| Abstainer | 46 (51.1) | 37 (66.1) | 9 (26.5) |  |
| 1–7 drinks/week | 39 (43.3) | 18 (32.1) | 21 (61.8) |  |
| 8–12 drinks/week | 2 (2.2) | 0 (0) | 2 (5.9) |  |
| ≥13 drinks/week | 3 (3.3) | 1 (1.8) | 2 (5.9) |  |
| Recent substance use since last visit[c] | 7 (7.8) | 4 (7.1) | 3 (8.8) | 1.0[a] |
| Use of prescribed psychotropic medication | 24 (26.7) | 18 (32.1) | 6 (17.7) | 0.13 |
| **Cardiometabolic factors** |  |  |  |  |
| Postmenopausal | 65 (72.2) | 44 (78.6) | 21 (61.8) | 0.084 |
| Body mass index (kg/m$^2$), median (IQR) | 32.6 (28.5–38.4) | 31.3 (28.3–38.1) | 35.0 (28.6–41.8) | 0.096 |
| **Psychosocial stress** |  |  |  |  |
| PSS-10[d], median (IQR) | 17 (12–22) | 16 (10–21) | 17 (13–24) | 0.17 |
| PCL-C[e], median (IQR) | 41 (26–52) | 40 (26–52) | 42.5 (25–57) | 0.45 |
| RSHS[f], median (IQR) | 18 (4–34) | 12.5 (2.5–31.5) | 21.5 (6–36) | 0.34 |
| Latent profile membership |  |  |  | 0.44 |
| Low psychosocial stress | 34 (37.8) | 24 (42.9) | 10 (29.4) |  |
| Moderate psychosocial stress | 40 (44.4) | 23 (41.1) | 17 (50.0) |  |
| High psychosocial stress | 16 (17.8) | 9 (16.1) | 7 (20.6) |  |

Abbreviations: IQR, interquartile range; PSS-10, PCL-C, PTSD Checklist-Civilian Version; Perceived Stress Scale; RSHS, Racialized Sexual Harassment Scale; WLWH, women living with HIV; col, column.

[a] P-values for race/ethnicity, current alcohol use, and recent substance use were obtained using Fisher's exact testing.

[b] Other race/ethnicity includes 1 American Indian/Alaska Native participant and 31 multiracial participants. Among multiracial participants, reported races included Black (n = 28), American Indian/Alaska Native (n = 24), white (n = 18), Asian (n = 4), and Native Hawaiian/Pacific Islander (n = 1) (non-mutually exclusive categories).

[c] Substance use includes use of cocaine (n = 2), crack/freebase cocaine (n = 5), heroin (n = 2), or methamphetamines (n = 1).

[d] The PSS-10 is a 10-item, 5-point Likert scale that measures perceived stress in the last month. Response options range from "never" (each assigned a score of 0) to "very often" (each assigned a score of 4) and four positively stated items are reverse-coded. The possible range of the total score is 0–40.

[e] The PCL-C is a 17-item, 5-point Likert scale that measures symptoms of posttraumatic stress disorder in the last month. Response options range from "not at all" (each assigned a score of 1) to "extremely" (each assigned a score of 5). The possible range of the total score is 17–85.

[f] The RSHS is a 21-item, 5-point Likert scale that measures experiences of race- and gender-based harassment in the last year. Response options range from "never" (each assigned a score of 0) to "very often" (each assigned a score of 4). The possible range of the total score is 0–84.

Although distributions of age, race/ethnicity, HIV serostatus, current alcohol use, and BMI were similar among participants in low, moderate, and high psychosocial stress latent profiles, women with high psychosocial stress were more likely than women with low psychosocial stress to not have completed high school (38% vs. 12%), have an income ≤$30,000 per year (94% vs. 44%), currently smoke (50% vs. 9%), report recent substance use (19% vs. 0%), and use a prescribed psychotropic medication (44% vs. 9%) (all p<0.05; data not shown).

## Psychosocial stress

Distributions of perceived stress, posttraumatic stress, experiences of race- and gender-based harassment, and latent profile membership were similar between WLWH and HIV-negative women (Table 3). These measures were also similar across racial/ethnic subgroups (all p>0.30), although there were small numbers of non-Hispanic white and Hispanic participants. Psychosocial stress measures were moderately correlated: ρ = 0.65 for the correlation between PSS-10 and PCL-C scores (p<0.0001); ρ = 0.35 for the correlation between PSS-10 and RSHS scores (p = 0.0008); and ρ = 0.46 for the correlation between PCL-C and RSHS scores (p<0.0001).

## Urine collection compliance and behaviors

The median duration between participants' study visit and the start of their 12-hour overnight urine collection was 1 day (IQR: 0–5; range: 0–19). The majority of participants (n = 83; 92%) reported collecting their urine every time they urinated during the 12-hour collection period, whereas the remaining seven participants reported missing exactly one void. An additional four participants (4%) reported that more than a few drops of urine were missing from the collection container due to another reason such as spilling. Aside from one reported asthma attack, no respiratory illnesses (e.g., cold, flu), fevers, or other acute illnesses were reported during the collection period. Three participants (3%) reported alcohol use (two reported one drink and one reported five drinks), five (6%) reported marijuana use, and one (1%) reported crack cocaine use on the day of the collection. Most (n = 75; 83%) reported waking up at least once during the night of the collection.

## Neuroendocrine biomarkers

Distributions of neuroendocrine biomarker concentrations stratified by HIV serostatus are provided in Fig 1 and Table 4. Based on distribution percentiles in the sample, high-risk quartiles were defined as ≥18.9 μg/g creatinine for urinary free cortisol, ≤35 μg/dL for serum DHEA-S, ≥3.7 μg/g creatinine for urinary epinephrine, and ≥43.9 μg/g creatinine for urinary norepinephrine. Overall, the value for the composite neuroendocrine biomarker index was 0 for 33 participants (37%), 1 for 32 participants (36%), 2 for 16 participants (18%), 3 for eight participants (9%), and 4 for one participant (1%). The distribution of the composite biomarker index was similar between WLWH and HIV-negative women (p = 0.47).

Regarding specific biomarkers, DHEA-S concentrations were lower among WLWH (median 47 μg/dL) compared with HIV-negative women (median 73 μg/dL; p = 0.0048), whereas distributions of cortisol, epinephrine, and norepinephrine were similar. After adjusting for light (β = 6.6; p = 0.52) and moderate/heavy (β = 58.9; p = 0.0066) alcohol use and psychotropic medication use (β = -21.4; p = 0.046), WLWH no longer had significantly lower serum DHEA-S concentrations compared with HIV-negative women (β = -11.0; 95% confidence interval [CI]: -31.3, 9.3; p = 0.29).

Biomarkers were not different across racial/ethnic subgroups (p = 0.12 for cortisol; p = 0.82 for epinephrine; p = 0.96 for norepinephrine; p = 0.60 for composite biomarker index),

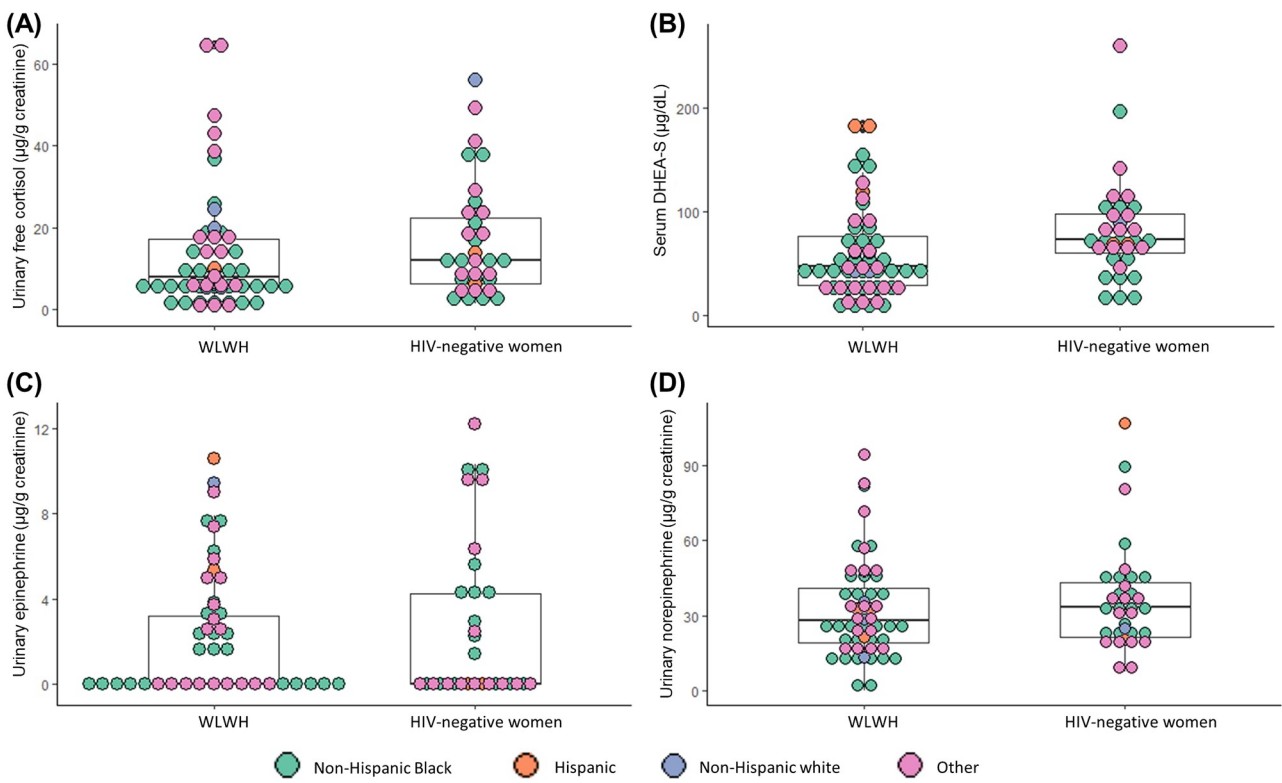

**Fig 1. Dot and box plots of concentrations of (A) urinary free cortisol, (B) serum DHEA-S, (C) urinary epinephrine, and (D) urinary norepinephrine by HIV serostatus (n = 90).** Laboratory test results below the lower limit of quantification were coded as 0 (n = 5 for urinary free cortisol, n = 52 for urinary epinephrine, and n = 1 for urinary norepinephrine). The central line, top border, and bottom border of box plots represent the median, 25th, and 75th percentiles, respectively, and the whiskers display 1.5 times the interquartile range. Abbreviations: DHEA-S, dehydroepiandrosterone sulfate; WLWH, women living with HIV.

although DHEA-S showed a trend toward significance: median 49 μg/dL (IQR: 32–77) among non-Hispanic Black women, median 57 μg/dL (IQR: 46–64) among non-Hispanic white women, median 119 μg/dL (IQR: 75–178) among Hispanic women, and median 64 μg/dL (IQR: 30–92) among multiracial or American Indian/Alaska Native women (p = 0.064).

## Association between latent profile and composite neuroendocrine biomarker index

Unadjusted and adjusted associations between psychosocial stress latent profile and the composite neuroendocrine biomarker index are presented in Table 5 for all women, WLWH, and HIV-negative women. Prior to covariate adjustment, women with moderate psychosocial stress had a 0.51-higher (95% CI: 0.06, 0.97) mean composite biomarker index compared with women with low psychosocial stress. Among WLWH, women with high psychosocial stress had a 0.85-higher (95% CI: 0.06, 1.64) mean composite biomarker index compared with women with low psychosocial stress. Associations for CD4+ T cell count and HIV viral load (detectable vs. undetectable using a 20 copies/mL threshold) with the composite biomarker index were not significant (p = 0.56 and p = 0.23, respectively). In full multivariable models, psychosocial stress was not significantly associated with the composite biomarker index among all women, WLWH, or HIV-negative women. Results based on parsimonious models were largely similar, although high (vs. low) psychosocial stress appeared marginally associated

**Table 4.  Distribution of neuroendocrine biomarkers by HIV serostatus (n = 90).**

| | Overall (n = 90) median (IQR) | WLWH (n = 56) median (IQR) | HIV-negative women (n = 34) median (IQR) | p |
|---|---|---|---|---|
| **HPA axis biomarkers** | | | | |
| Cortisol (μg/g creatinine) | 9.3 (5.7–18.9) | 7.8 (5.1–17.1) | 12.0 (6.3–22.8) | 0.12 |
| DHEA-S (μg/dL) | 58 (35–89) | 47 (28–76.5) | 73 (59–99) | 0.0048 |
| **SNS biomarkers** | | | | |
| Epinephrine (μg/g creatinine)[a] | UD (UD-3.7) | UD (UD-3.3) | UD (UD-4.3) | 0.83 |
| Norepinephrine (μg/g creatinine) | 31.1 (21.1–43.9) | 28.2 (19.3–41.9) | 33.3 (21.3–43.9) | 0.40 |
| | n (%) | n (col %) | n (col %) | p |
| **Composite neuroendocrine biomarker index[b]** | | | | 0.47[c] |
| 0 | 33 (36.7) | 22 (39.3) | 11 (32.4) | |
| 1 | 32 (35.6) | 17 (30.4) | 15 (44.1) | |
| 2 | 16 (17.8) | 12 (21.4) | 4 (11.8) | |
| 3 | 8 (8.9) | 4 (7.1) | 4 (11.8) | |
| 4 | 1 (1.1) | 1 (1.8) | 0 (0) | |

Abbreviations: DHEA-S, dehydroepiandrosterone sulfate; HPA, hypothalamic pituitary adrenal; IQR, interquartile range; SNS, sympathetic nervous system; UD, undetectable; WLWH, women living with HIV; col, column.

[a] Fifty-two participants (58%) had epinephrine concentrations less than the lower limit of quantification (i.e., <2 μg per L urine).

[b] Calculated as the number of the four neuroendocrine biomarkers in the highest risk quartile (i.e., the upper quartile for cortisol, epinephrine, and norepinephrine, and the lower quartile for DHEA-S).

[c] The p-value for the composite neuroendocrine biomarker index was obtained using Fisher's exact testing.

**Table 5.  Association between psychosocial stress and composite neuroendocrine biomarker index among all women and by HIV serostatus (n = 90).**

| | Unadjusted model β (95% CI) | p | Full model [a] β (95% CI) | p | Parsimonious Model [b] β (95% CI) | p |
|---|---|---|---|---|---|---|
| **Overall (n = 90)** | | | | | | |
| Psychosocial stress latent profile membership | | | | | | |
| Low | ref | | ref | | ref | |
| Moderate | **0.51 (0.06, 0.97)** | **0.028** | 0.31 (-0.25, 0.88) | 0.27 | 0.38 (-0.12, 0.87) | 0.13 |
| High | 0.33 (-0.27, 0.92) | 0.28 | 0.31 (-0.45, 1.08) | 0.41 | 0.38 (-0.29, 1.05) | 0.26 |
| **Among WLWH (n = 56)** | | | | | | |
| Psychosocial stress latent profile membership | | | | | | |
| Low | ref | | ref | | ref | |
| Moderate | 0.42 (-0.17, 1.01) | 0.16 | 0.08 (-0.82, 0.99) | 0.85 | 0.29 (-0.36, 0.94) | 0.26 |
| High | **0.85 (0.06, 1.64)** | **0.036** | 0.66 (-0.47, 1.78) | 0.24 | 0.80 (-0.004, 1.61) | 0.051 |
| **Among HIV-negative women (n = 34)** | | | | | | |
| Psychosocial stress latent profile membership | | | | | | |
| Low | ref | | ref | | ref | |
| Moderate | 0.61 (-0.12, 1.35) | 0.10 | 0.61 (-0.47, 1.68) | 0.25 | 0.64 (-0.19, 1.47) | 0.13 |
| High | -0.37 (-1.28, 0.54) | 0.41 | -0.07 (-1.71, 1.58) | 0.93 | -0.21 (-1.15, 0.73) | 0.65 |

Abbreviations: CI, confidence interval; WLWH, women living with HIV; ref, reference group.

Bolded numeric values indicate p<0.05.

[a] Adjusted for HIV status (only in the overall sample), age, race/ethnicity, education at study entry, income, current smoking, alcohol use, recent substance use, use of a prescribed psychotropic medication, body mass index, menopausal status, CD4+ T cell count (only among WLWH), and HIV viral load (only among WLWH).

[b] In the overall sample, adjusted for HIV status, age, race/ethnicity, education at study entry, income, alcohol use, recent, substance use, and body mass index. Among WLWH, adjusted for age, race/ethnicity, income, current smoking, and body mass index. Among HIV-negative women, adjusted for age, race/ethnicity, education at study entry, current smoking, and recent substance use.

**Table 6. Association between psychosocial stress and composite neuroendocrine biomarker index among monoracial non-Hispanic Black women overall and by HIV serostatus (n = 48).**

|  | Unadjusted model | | Full model [a] | | Parsimonious Model [b] | |
|---|---|---|---|---|---|---|
|  | β (95% CI) | p | β (95% CI) | p | β (95% CI) | p |
| **Overall (n = 48)** |  |  |  |  |  |  |
| Psychosocial stress latent profile membership |  |  |  |  |  |  |
| Low | ref |  | ref |  | ref |  |
| Moderate | 0.53 (-0.06, 1.13) | 0.078 | 0.30 (-0.42, 1.03) | 0.40 | 0.33 (-0.24, 0.90) | 0.25 |
| High | **0.73 (-0.11, 1.57)** | **0.088** | **1.32 (0.20, 2.43)** | **0.022** | **1.31 (0.41, 2.21)** | **0.0053** |
| **Among WLWH (n = 31)** |  |  |  |  |  |  |
| Psychosocial stress latent profile membership |  |  |  |  |  |  |
| Low | ref |  | ref |  | ref |  |
| Moderate | 0.37 (-0.35, 1.09) | 0.30 | 0.13 (-1.41, 1.68) | 0.86 | 0.06 (-0.76, 0.88) | 0.88 |
| High | **0.98 (0.01, 1.96)** | **0.049** | **1.93 (0.02, 3.83)** | **0.048** | **1.64 (0.54, 2.73)** | **0.0052** |
| **Among HIV-negative women (n = 17)** |  |  |  |  |  |  |
| Psychosocial stress latent profile membership |  |  |  |  |  |  |
| Low | ref |  | ref |  | ref |  |
| Moderate | 0.72 (-0.37, 1.81) | 0.18 | 0.36 (-0.66, 1.38) | 0.38 | 0.32 (-0.41, 1.06) | 0.32 |
| High | 0.17 (-1.52, 1.86) | 0.84 | **2.54 (0.41, 4.67)** | **0.030** | **2.51 (0.93, 4.08)** | **0.0080** |

Abbreviations: CI, confidence interval; WLWH, women living with HIV; ref, reference group.

Bolded numeric values indicate p<0.05.

[a] Adjusted for HIV status (only in the overall sample), age, education at study entry, income, current smoking, alcohol use, recent substance use, use of a prescribed psychotropic medication, body mass index, menopausal status, CD4+ T cell count (only among WLWH), and HIV viral load (only among WLWH).

[b] In the overall sample, adjusted for HIV status, age, income, alcohol use, recent substance use, and body mass index. Among WLWH, adjusted for age, recent substance use, use of a prescribed psychotropic medication, and HIV viral load. Among HIV-negative women, adjusted for age, education at study entry, current smoking, alcohol use, recent substance use, body mass index, and menopausal status.

with higher mean composite biomarker index values among WLWH (β = 0.80; 95% CI: -0.004, 1.61).

Unadjusted and adjusted associations among the subgroup of non-Hispanic Black women identifying exclusively as Black (and not another race) are presented in Table 6. Prior to covariate adjustment, monoracial Black WLWH with high psychosocial stress had a 0.98-higher (95% CI: 0.01, 1.96) mean composite biomarker index compared with monoracial Black WLWH with low psychosocial stress. In full multivariable models, high (vs. low) psychosocial stress was positively and significantly associated with higher mean composite biomarker index values among all monoracial Black women (β = 1.32; 95% CI: 0.20, 2.43), monoracial Black WLWH (β = 1.93; 95% CI: 0.02, 3.83), and monoracial Black HIV-negative women (β = 2.54; 95% CI: 0.41, 4.67). Results based on parsimonious models were similar. Associations were attenuated and none remained significant in analyses among all non-Hispanic Black women, which included those who identified exclusively as Black plus those who also identified with another race (data not shown).

## Sensitivity analyses

Results were similar after excluding 13 participants who reported either not collecting their urine every time they urinated during the collection period, losing more than a few drops of urine from the collection container due to spilling, or heavy alcohol use (i.e., 5 drinks) or crack cocaine use on the day of the collection. In multivariable models, psychosocial stress was not significantly associated with the composite biomarker index among all women, all WLWH, or

all HIV-negative women. Among monoracial non-Hispanic Black women, high (vs. low) psychosocial stress remained positively and significantly associated with higher mean composite biomarker index values based on results from full multivariable models: β = 1.41 (95% CI: 0.23, 2.59; p = 0.021) among all Black women, β = 2.17 (95% CI: 0.16, 4.17; p = 0.037) among Black WLWH, and β = 2.25 (95% CI: 1.37, 3.13; p = 0.0038) among Black HIV-negative women.

After using a bidirectional cutoff to define the high-risk quartile for urinary free cortisol (i.e., ≤2.9 μg/g creatinine or ≥29.1 μg/g creatinine), the fully adjusted association between latent profile and the composite biomarker index remained non-significant for all women, all WLWH, and all HIV-negative women. The fully adjusted association between high (vs. low) psychosocial stress and the composite biomarker index was attenuated among all monoracial Black women (β = 1.11; 95% CI: -0.17, 2.38; p = 0.086), whereas it changed little among monoracial Black WLWH (β = 1.88; 95% CI: 0, 3.76; p = 0.050) and monoracial Black HIV-negative women (β = 2.70; 95% CI: 0.23, 5.17; p = 0.039).

## Discussion

In this study, we found a null association between psychosocial stress and a composite neuroendocrine biomarker index among the overall sample of WLWH and HIV-negative women and did not find evidence of a difference in this association between WLWH and HIV-negative women. However, our data demonstrate that higher levels of self-reported psychosocial stress were associated with more extreme concentrations of neuroendocrine biomarkers specifically among Black WLWH and HIV-negative women, although this was only the case among monoracial Black participants (and not among all Black participants, including those who identified with at least one other race). By incorporating multiple measures of psychosocial stress (i.e., general perceived stress, posttraumatic stress, and experiences of race- and gender-based harassment), we were able to more fully capture different domains of stress experienced by women. Given the strength of the association detected among subgroups of Black WLWH and HIV-negative women, findings among Black women reached statistical significance even with a relatively small sample size. Although this cross-sectional study cannot establish temporality or causality, our findings provide some support that psychosocial stress promotes hyperactivity of the HPA axis and SNS among Black WLWH and those who are HIV-negative, and highlights a potential mechanism by which chronic stress could contribute to increased inflammation and metabolic or cardiovascular abnormalities over time. Importantly, in this study, psychosocial stress appeared to be relevant in these neural pathways irrespective of HIV status. Additional studies are needed to further distinguish the relative contributions of psychosocial stressors from the effects of chronic HIV infection.

This study builds upon prior research examining correlates of composite biomarker indices based on neuroendocrine, inflammatory, metabolic, and/or cardiovascular markers among other subpopulations of PLWH [53–57]. Previously identified correlates of having greater composite biomarker index values have included HIV-positive status among Black and Latina mothers [54], lower psychological resilience among older Black PLWH [56], PTSD among Black bisexual men living with HIV [55], and adverse childhood experiences among middle-aged PLWH [53]. We specifically focused on neuroendocrine biomarkers, which are proximal markers of HPA axis and SNS activation that influence inflammation and metabolic and cardiovascular dysfunction [28, 58, 59]. Unlike prior similar studies, we measured an intersectional form of stress experienced by women of color (i.e., race- and gender-related).

Although results were null for the larger sample of women of all races and ethnicities, our findings for Black WLWH and HIV-negative women are consistent with prior research of

discrimination and neuroendocrine activity among Black populations. The association between racial discrimination and HPA axis dysregulation due to repeated and prolonged cortisol secretion is well documented in Black Americans [77–79]. Greater racial discrimination has also been linked to increased catecholamine activity among Black men and women [80, 81]. Importantly, some evidence suggests that protective factors such as higher educational attainment, higher income, and greater religious engagement can buffer the association between discrimination and neuroendocrine activity among Black Americans [82, 83].

Although we assessed recent experiences of race- and gender-related harassment, our use of general symptom-based measures for perceived stress and posttraumatic stress limited our ability to identify the sources of those types of stress reported by participants. Previously reported stressors among WLWH have included difficulties with HIV serostatus disclosure, challenges with medication adherence, HIV-related discrimination, mental health or substance use problems, relationship challenges, caretaking for family members, and financial difficulties [84]. Null findings among the overall study population may have been related to the use of general measures for perceived and posttraumatic stress. Such general forms of stress may have had limited influence on the neuroendocrine biomarkers evaluated in this study.

Results should be interpreted within the context of several limitations. First, a 12-hour overnight urine collection period was chosen because it was more practical and acceptable for study participants compared with a full 24-hour collection period. Thus, biomarker measures correspond to the nadir of HPA axis and SNS activity and do not reflect the full diurnal cycle of cortisol, norepinephrine, and epinephrine release. Second, sample sizes were relatively small in analyses stratified by HIV status and/or restricted to non-Hispanic Black participants, which resulted in less precise estimates for associations. There were too few white and Hispanic participants to meaningfully conduct analyses among those racial/ethnic subgroups. Third, we used data-driven methods to classify psychosocial stress as low, moderate, or high (i.e., latent profile analysis) and to define high-risk ranges for each biomarker (i.e., upper or lower quartiles). While this approach facilitated comparisons of composite biomarker index values according to varying degrees of stress levels within the sample, it limits our ability to determine whether stress levels and biomarker concentrations were clinically meaningful. This initial study demonstrated feasibility and additional research should evaluate the association between levels of stress and clinical endpoints. Fourth, covariates were measured at core WIHS visits that occurred up to nine months before the date of data collection for this study. The fact that values for time-varying covariates could have changed between when they were measured and when psychosocial stress and neuroendocrine biomarkers were measured could have resulted in residual confounding. Fifth, we lacked data on relevant behavioral factors such as regular diet, exercise, and sleep habits, which could have resulted in unmeasured confounding.

## Conclusions

In summary, this study found that higher levels of psychosocial stress were associated with more extreme concentrations of neuroendocrine biomarkers among monoracial Black women living with or without HIV, but not among the larger study population. Findings point to a plausible mechanism by which chronic stress could contribute to increased inflammation and metabolic or cardiovascular abnormalities over time among Black WLWH and HIV-negative women. Future research with larger study samples should evaluate whether this association differs by HIV status among Black women, by race or ethnicity among WLWH, or by gender among PLWH. The effects of psychosocial stress and neuroendocrine biomarker concentrations on inflammatory, metabolic, and cardiovascular outcomes among WLWH should also

be further evaluated. Finally, research is needed to determine if interventions to manage or reduce psychosocial stress can influence clinical outcomes among Black and other WLWH.

## Acknowledgments

We would like to thank Margo Daniel, Dao Mai, Kimberly Bennett, Millicent Rawle, Kathleen Digilio, Mable Torre, Sheree Bailey-Johnson, Gloria Woodfork, Cuiwei Wang, and all study participants for their important contributions to this research project. Data in this manuscript were collected by the Women's Interagency HIV Study (WIHS), now the MACS/WIHS Combined Cohort Study (MWCCS). MWCCS (Principal Investigators): Data Analysis and Coordination Center (Gypsyamber D'Souza, Stephen Gange, and Elizabeth Golub); and Metropolitan Washington Clinical Research Site (Seble Kassaye and Daniel Merenstein). The contents of this publication are solely the responsibility of the authors and do not represent the official views of the National Institutes of Health (NIH).

## Author Contributions

**Conceptualization:** Matthew E. Levy, Sabyasachi Sen, Amanda D. Castel, Michael Plankey, Sherry Molock, Federico Asch, Lakshmi Goparaju, Seble Kassaye.

**Formal analysis:** Matthew E. Levy, Ansley Waters, Sabyasachi Sen, Amanda D. Castel, Michael Plankey, Sherry Molock, Federico Asch, Lakshmi Goparaju, Seble Kassaye.

**Funding acquisition:** Matthew E. Levy, Sabyasachi Sen, Amanda D. Castel, Michael Plankey, Sherry Molock, Federico Asch, Lakshmi Goparaju, Seble Kassaye.

**Investigation:** Matthew E. Levy, Ansley Waters, Sabyasachi Sen, Amanda D. Castel, Michael Plankey, Sherry Molock, Federico Asch, Lakshmi Goparaju, Seble Kassaye.

**Project administration:** Matthew E. Levy, Lakshmi Goparaju, Seble Kassaye.

**Writing – original draft:** Matthew E. Levy, Ansley Waters.

**Writing – review & editing:** Matthew E. Levy, Ansley Waters, Sabyasachi Sen, Amanda D. Castel, Michael Plankey, Sherry Molock, Federico Asch, Lakshmi Goparaju, Seble Kassaye.

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
