## [Decision Letter · Decision Letter 0]

26 Feb 2021

PONE-D-20-27314

Psychosocial stress is associated with neuroendocrine biomarker concentrations among Black women living with or without HIV

PLOS ONE

Dear Dr. Levy,

Thank you for submitting your manuscript to PLOS ONE. After careful consideration, we feel that it has merit but does not fully meet PLOS ONE’s publication criteria as it currently stands. Therefore, we invite you to submit a revised version of the manuscript that addresses the points raised during the review process.

We look forward to receiving your revised manuscript.

Kind regards,

Yukiko Washio, Ph.D.

Academic Editor

PLOS ONE

Journal Requirements:

We note that one or more of the authors are employed by a commercial company: Westat.

(2) Please also provide an updated Competing Interests Statement declaring this commercial affiliation along with any other relevant declarations relating to employment, consultancy, patents, products in development, or marketed products, etc.  

Reviewers' comments:

Reviewer's Responses to Questions

**Comments to the Author**

1. Is the manuscript technically sound, and do the data support the conclusions?

Reviewer #1: Yes

Reviewer #2: Yes

2. Has the statistical analysis been performed appropriately and rigorously? 

Reviewer #1: Yes

Reviewer #2: No

3. Have the authors made all data underlying the findings in their manuscript fully available?

Reviewer #1: Yes

Reviewer #2: No

4. Is the manuscript presented in an intelligible fashion and written in standard English?

Reviewer #1: Yes

Reviewer #2: Yes

5. Review Comments to the Author

Reviewer #1: “Psychosocial stress is associated with neuroendocrine biomarker concentrations

among Black women living with or without HI" is a paper whose topic may be of interest to the readership of PLOS ONE. The current study takes an intersectional approach to examine the association between self-reported psychosocial stress and neuroendocrine biomarkers of stress among women living with HIV, with an emphasis on black women. This study contributes to the literature by focusing on understanding the potential mechanisms of cardiovascular disease among cis-gender women living with HIV, a population that has been neglected in research to date. This study supports and extends previous research by highlighting the role of stress, especially racial and gender-based harassment on the physiological functioning of black women living with HIV. However, I have one major concern about the misalignment between how the article is framed in the introduction and the results that are presented. What follows are my concerns and points of clarification.

Overall Concern

The authors frame the manuscript as an examination of stress in Black women living with HIV. This is present in the title and introduction. However, the authors actually examined the role of self-reported stress and biomarkers of stress among women of all ethnicities. As a result of this misalignment, the point of the paper gets lost. It appears the original research question was related to all women living with HIV as compared to those who were HIV-negative and that the sub-analysis among black women was part of a fishing expedition given the null findings of the main analysis.

Given the authors interest in understanding how race and gender-related stress impact health of women living with HIV, I recommend that the authors dedicate some time to fully addressing how racism and sexism may impact the stress and health of Black women living with HIV in the introduction; remove the analysis which included all women (currently the main analysis), and focus on the analysis examining the associations of interests among Black women living with HIV and those who were not. Alternatively, the authors might keep the analysis which includes all women as and include it with the other supplementary analysis that was conducted – however, I believe it would be best just to remove it.

Introduction

1. I would be careful when addressing how stress related to socioeconomic status is related to the health of women living with HIV. The authors do not measure socioeconomic status is a meaningful way (above or below $30,000 is not an approximate proxy for SES), therefore, the introduction should focus on the potential impact of racism, sexism, posttraumatic stress, and other perceived stress on the health of this population.

Methods

1. It should be noted that no traumatic event was indexed on the PTSD checklist. This is problematic because the “Stressful event” referenced in the questions may not meet criteria for a traumatic event. This is briefly noted in the limitations but should also be stated in the methods. The authors may also consider removing this measure all together since it is very difficult to differentiate between generalized anxiety and PTSD with understanding the index event. For example, the sample questions presented by the authors is “Avoiding activities or situations because they reminded you of a stressful experience”. It is difficult to distinguish between social anxiety and re-experiencing a traumatic event if there is no index event. Alternatively, the authors may separate the scores for racial and gender harassment and use them as separate indicators in the LCA. Therefore, they could still have three indicators of the LCA if they removed the PTSD scale.

2. Please clarify that the subgroup analysis conducted among Black women is stratified by HIV status.

Discussion

1. Again, the authors should be careful with attributing the stress of participants to SES, since SES was not appropriately measured and not included in the measurement of stress.

2. Also, the authors should be careful not to attributed PTSD symptoms to the experiences of violence reported by the participants, since no index event was reported for the checklist.

Decision: Revise and Resubmit, with Major Revisions

Reviewer #2: The focus of the study was to examine the relationship between psychosocial stress and neuroendocrine biomarkers among women living with HIV. This article was well written and presents novel findings about a population which experience marginalization due to race, gender, and HIV status. The major concerns about this article are how the findings are framed and the choice in analytic methods. I offer a few suggestions for improving the manuscript.

Major comments

- The premise of the article is that racial discrimination could lead to differential outcomes among Black women, however the stress measures were general. For the one measure that specifically identified race-related stress, the racial and gender harassment scale, participants across racial groups had similar scores. Additionally, biomarker scores were similar across group. Temper findings and lead with the null finding that the relation between psychosocial stress latent profile and biomarker index was null, however there is some evidence that this relationship varies by racial category. In the discussion explain why the main finding and subgroup analyses among WLWH compared to HIV-negative women, were null.

- The sub-group analyses within Black women are rather small and do not provide anything meaningful beyond what is gleaned from the full sample of Black women. Cut the analyses for the WLWH and HIV negative women.

- Linear regression was used for multivariable analyses, however the biomarker index is not truly a continuous variable. Ordinal logistic regression should be used instead.

Minor comments

- Page 11, Line 219: If possible, find an alternate item to use as the example that does not include the racial slurs.

- Page 11, Line 224: Describe race/ethnicity categories and rationale for all the participants who were grouped in the other category. Were sensitivity analyses conducted to examine whether including multiracial participants who checked Black into the analyses for participants who solely checked Black? Often multiracial people who are Black and another race are racialized as Black and may have similar stressors.

6. PLOS authors have the option to publish the peer review history of their article (what does this mean?). If published, this will include your full peer review and any attached files.

Reviewer #1: No

Reviewer #2: No

---

## [Author Response · Author response to Decision Letter 0]

23 Aug 2021

Dear Dr. Washio,

Thank you for the opportunity to revise this manuscript, titled “Psychosocial stress and neuroendocrine biomarker concentrations among women living with or without HIV” (PONE-D-20-27314). Please find below indication of where changes were made in response to the comments provided by the reviewers. We are happy to make additional revisions as deemed necessary.

Reviewer 1

Comment: “Psychosocial stress is associated with neuroendocrine biomarker concentrations among Black women living with or without HIV" is a paper whose topic may be of interest to the readership of PLOS ONE. The current study takes an intersectional approach to examine the association between self-reported psychosocial stress and neuroendocrine biomarkers of stress among women living with HIV, with an emphasis on black women. This study contributes to the literature by focusing on understanding the potential mechanisms of cardiovascular disease among cis-gender women living with HIV, a population that has been neglected in research to date. This study supports and extends previous research by highlighting the role of stress, especially racial and gender-based harassment on the physiological functioning of black women living with HIV. However, I have one major concern about the misalignment between how the article is framed in the introduction and the results that are presented. What follows are my concerns and points of clarification.

Response: We thank Reviewer 1 for reviewing this manuscript and for providing comprehensive and thoughtful feedback. We have reframed the manuscript and provided additional information to address this reviewer’s questions and concerns. Please see our responses below each specific comment for additional details regarding our revisions, including those pertaining to the major concern about the misalignment between how the article is framed and the introduction and results that are presented.

Comment: Overall Concern: The authors frame the manuscript as an examination of stress in Black women living with HIV. This is present in the title and introduction. However, the authors actually examined the role of self-reported stress and biomarkers of stress among women of all ethnicities. As a result of this misalignment, the point of the paper gets lost. It appears the original research question was related to all women living with HIV as compared to those who were HIV-negative and that the sub-analysis among black women was part of a fishing expedition given the null findings of the main analysis.

Given the authors interest in understanding how race and gender-related stress impact health of women living with HIV, I recommend that the authors dedicate some time to fully addressing how racism and sexism may impact the stress and health of Black women living with HIV in the introduction; remove the analysis which included all women (currently the main analysis), and focus on the analysis examining the associations of interests among Black women living with HIV and those who were not. Alternatively, the authors might keep the analysis which includes all women as and include it with the other supplementary analysis that was conducted – however, I believe it would be best just to remove it.

Response: We thank the reviewer for these comments. As pre-planned, our primary objective was to “evaluate associations between psychosocial stress (i.e., perceived stress, posttraumatic stress, and experiences of race- and gender-based harassment) and a composite neuroendocrine biomarker index among WLWH and HIV-negative women” of all races and ethnicities (line 102 in the version with tracked changes). We “hypothesized that this association would be stronger among WLWH compared with HIV-negative women” (line 107 in the version with tracked changes). Rather than representing a fishing expedition, our analyses conducted among the subgroup of Black participants were pre-planned based on (1) the large proportion of WLWH in the United States, DC, and this study population who identify as Black; (2) the body of empirical research demonstrating the relationship between racism or racial microaggressions and stress; and (3) our inclusion of the Racialized Sexual Harassment Scale as a measure of harassment related to one’s race or ethnicity and/or gender, which has greater relevance to women of color. We have clarified in the Introduction that this subgroup analysis was secondary: “Given that Black WLWH disproportionately experience stress related to intersecting social-structural vulnerabilities [60-62], we also conducted a secondary subgroup analysis in which we evaluated this association specifically among Black women” (line108 in the version with tracked changes). We have reframed this manuscript to more accurately reflect our original objectives and have revised the title to “Psychosocial stress and neuroendocrine biomarker concentrations among women living with or without HIV.” Reflecting these objectives, we have opted to retain the overall study population (of all races or ethnicities) in our analysis and to continue to report both results for the overall study sample and the subset of Black participants.

The introduction is now better aligned with this focus. The second paragraph focuses on psychosocial stress among WLWH in general, and we further expand upon this paragraph in the third paragraph to address how racism and sexism may impact the stress and health of Black women and Black WLWH, consistent with our secondary objective: “At the end of 2019, 57% of all WLWH in the United States were Black women [20]. Among Black women, discrimination attributed to racism and sexism is common and is associated with elevated stress [21, 22]. Gendered racism, or the simultaneous experience of both racism and sexism, is an inherent stressor that can negatively affect personal relationships, professional and economic opportunities, and mental and physical health outcomes among Black women [23-26]. Gendered racism causes psychosocial stress both directly and through increased vulnerability to other stressors [21, 23-25]. Further, among Black WLWH, the effects of gendered racism and HIV-based discrimination can compound one another, representing key social-structural barriers to HIV care [27].”

Our discussion section is also better aligned with this focus, as it begins with: “In this study, we found a null association between psychosocial stress and a composite neuroendocrine biomarker index among the overall sample of WLWH and HIV-negative women and did not find evidence of a difference in this association between WLWH and HIV-negative women. However, our data demonstrate that higher levels of self-reported psychosocial stress were associated with more extreme concentrations of neuroendocrine biomarkers specifically among Black WLWH and HIV-negative women, although this was only the case among monoracial Black participants (and not among all Black participants, including those who identified with at least one other race).”

Comment: Introduction: 1. I would be careful when addressing how stress related to socioeconomic status is related to the health of women living with HIV. The authors do not measure socioeconomic status is a meaningful way (above or below $30,000 is not an approximate proxy for SES), therefore, the introduction should focus on the potential impact of racism, sexism, posttraumatic stress, and other perceived stress on the health of this population.

Response: We agree with this comment and have removed mention of socioeconomic status from the introduction. The introduction focuses on the potential impact of perceived stress, posttraumatic stress, racism, and sexism, reflecting the key measures from this study.

Comment: Methods: 1. It should be noted that no traumatic event was indexed on the PTSD checklist. This is problematic because the “Stressful event” referenced in the questions may not meet criteria for a traumatic event. This is briefly noted in the limitations but should also be stated in the methods. The authors may also consider removing this measure all together since it is very difficult to differentiate between generalized anxiety and PTSD with understanding the index event. For example, the sample questions presented by the authors is “Avoiding activities or situations because they reminded you of a stressful experience”. It is difficult to distinguish between social anxiety and re-experiencing a traumatic event if there is no index event. Alternatively, the authors may separate the scores for racial and gender harassment and use them as separate indicators in the LCA. Therefore, they could still have three indicators of the LCA if they removed the PTSD scale.

Response: We thank the reviewer for raising this point and have added language to the methods section indicating that no traumatic event was indexed on the PTSD checklist: “The PCL-C asks about symptoms in relation to generic stressful experiences rather than in relation to a specific index event. Although this aspect of the PCL-C facilitates its use in different populations, it also limits its ability to identify sources of stress and determine whether stressful experiences meet criteria for a traumatic event” (line 206 in the version with tracked changes). We also acknowledged this limitation in the discussion section: “Although we assessed recent experiences of race- and gender-related harassment, our use of general symptom-based measures for perceived stress and posttraumatic stress limited our ability to identify the sources of those types of stress reported by participants” (line 556 in the version with tracked changes). Despite this limitation, we think that stress in relation to stressful experiences in general is of relevance to our research question and that there is value in retaining this measure in the analysis, even if PCL-C items were not assessed with respect to a specific traumatic event. We opted to retain the PCL-C measure in our analysis as planned.

Comment: 2. Please clarify that the subgroup analysis conducted among Black women is stratified by HIV status.

Response: We have clarified that the subgroup analysis conducted among Black women was stratified by HIV status: “Regression analyses were further conducted separately among all non-Hispanic Black participants, non-Hispanic Black WLWH, and non-Hispanic Black HIV-negative women” (line 329 in the version with tracked changes).

Comment: Discussion: 1. Again, the authors should be careful with attributing the stress of participants to SES, since SES was not appropriately measured and not included in the measurement of stress.

Response: We agree with this comment and have removed the previous text that attributed stress of participants to SES (previously, “With approximately two-thirds of the sample having had an annual income <$30,000 and approximately half having had at most a high school education, it is likely that challenges associated with low socioeconomic status contributed to stressors experienced by women in this study.”).

Comment: 2. Also, the authors should be careful not to attributed PTSD symptoms to the experiences of violence reported by the participants, since no index event was reported for the checklist.

Response: We agree with this comment and have removed the previous text that attributed PTSD symptoms to experiences of violence reported by the participants (previously, “Regarding possible sources of posttraumatic stress, 51% of WLWH and 55% of HIV-negative women in the WIHS have experienced physical violence, and 38% of WLWH and 40% of HIV-negative women in the WIHS have experienced sexual abuse.”).

Reviewer 2

Comment: The focus of the study was to examine the relationship between psychosocial stress and neuroendocrine biomarkers among women living with HIV. This article was well written and presents novel findings about a population which experience marginalization due to race, gender, and HIV status. The major concerns about this article are how the findings are framed and the choice in analytic methods. I offer a few suggestions for improving the manuscript.

Response: We thank Reviewer 2 for reviewing this manuscript and for providing comprehensive and thoughtful feedback. We have reframed the manuscript and provided additional information to address this reviewer’s questions and concerns. Please see our responses below each specific comment for additional details regarding our revisions, including those pertaining to how the findings are framed and the choice in analytic methods.

Comment: Major comments: The premise of the article is that racial discrimination could lead to differential outcomes among Black women, however the stress measures were general. For the one measure that specifically identified race-related stress, the racial and gender harassment scale, participants across racial groups had similar scores. Additionally, biomarker scores were similar across group. Temper findings and lead with the null finding that the relation between psychosocial stress latent profile and biomarker index was null, however there is some evidence that this relationship varies by racial category. In the discussion explain why the main finding and subgroup analyses among WLWH compared to HIV-negative women, were null.

Response: We thank the reviewer for these comments. We have reframed this manuscript to more accurately reflect our original objectives and have revised the title to “Psychosocial stress and neuroendocrine biomarker concentrations among women living with or without HIV.” We have tempered findings and now lead with the null finding; at the beginning of the Discussion section, we state: “In this study, we found a null association between psychosocial stress and a composite neuroendocrine biomarker index among the overall sample of WLWH and HIV-negative women and did not find evidence of a difference in this association between WLWH and HIV-negative women. However, our data demonstrate that higher levels of self-reported psychosocial stress were associated with more extreme concentrations of neuroendocrine biomarkers specifically among Black WLWH and HIV-negative women, although this was only the case among monoracial Black participants (and not among all Black participants, including those who identified with at least one other race).” We also provide a possible reason for why the main findings were null: “Null findings among the overall study population may have been related to the use of general measures for perceived and posttraumatic stress. Such general forms of stress may have had limited influence on the neuroendocrine biomarkers evaluated in this study” (line 587 in the version with tracked changes). It may also be related to the smaller sample size, which limited our ability to detect significant differences, unless quite large. As mentioned in the Conclusion paragraph, “Future research with larger study samples should evaluate whether this association differs by HIV status among Black women, by race or ethnicity among WLWH, or by gender among PLWH” (line 623 in the version with tracked changes). Regarding the finding that self-reported stress measures and biomarker scores did not appear different across racial and ethnic groups, there were only 5 non-Hispanic white participants and 5 Hispanic participants (and 31 biracial, mixed-race, or multiracial participants who identified with different combinations of races), limiting our capacity to detect differences by race or ethnicity in this study.

Comment: The sub-group analyses within Black women are rather small and do not provide anything meaningful beyond what is gleaned from the full sample of Black women. Cut the analyses for the WLWH and HIV negative women.

Response: In the introduction section, we have specified that “we hypothesized that this association would be stronger among WLWH compared with HIV-negative women” (line 107 in the version with tracked changes). Although the sample sizes for WLWH and HIV-negative women are small when restricted to Black participants, we have opted to retain these findings in the manuscript because a positive association was consistently found in separate analyses among all Black women, Black WLWH, and Black HIV-negative women. Each regression model converged and the association between high psychosocial stress and a higher neuroendocrine biomarker index score was significant in each of these subgroups. Although we would have likely been unable to detect a statistically significant difference in this association between WLWH and HIV-negative women, a reader may want to know whether there was at least some evidence of this association differing by HIV status, which we did not find. In the discussion section, we state this limitation that “sample sizes were relatively small in analyses stratified by HIV status and/or restricted to non-Hispanic Black participants, which resulted in less precise estimates for associations” (line 600 in the version with tracked changes).

Comment: Linear regression was used for multivariable analyses, however the biomarker index is not truly a continuous variable. Ordinal logistic regression should be used instead.

Response: We thank the reviewer for raising this point. Although we agree that ordinal logistic regression is an alternative choice, we believe that linear regression is also appropriate, particularly in the context of a smaller sample size. We have provided our rationale in the manuscript: “Although ordinal regression is a possible alternative, because the difference between the scores of 0 and 1, 1 and 2, 2 and 3, and 3 and 4 have the same interpretation (i.e., one additional biomarker in the high-risk range), we were able to appropriately harness the greater statistical power of linear regression for this analysis. This increased power is particularly useful in this study with a smaller sample size. This approach has been applied successfully elsewhere [72-76]" (line 315).

In the following similar studies, linear regression was used successfully to model allostatic load, the composite biomarker index score on which our study’s outcome measure was based:

1. Van Dyke ME, Baumhofer NK, Slopen N, Mujahid MS, Clark CR, Williams DR, et al. Pervasive discrimination and allostatic load in African American and white adults. Psychosom Med. 2020;82: 316-323.

2. Currie CL, Copeland JL, Metz GA, Chief Moon-Riley K, Davies CM. Past-year racial discrimination and allostatic load among indigenous adults in Canada: the role of cultural continuity. Psychosom Med. 2020;82: 99-107.

3. Freire A do NF, Barbosa JF de S, Pereira DS, Gomes CDS, Guerra RO. Allostatic load and stress biomarkers in a sample of community-dwelling older adults. Arch Gerontol Geriatr. 2020;87: 104006.

4. Hux VJ, Catov JM, Roberts JM. Allostatic load in women with a history of low birth weight infants: the National Health and Nutrition Examination Survey. J Womens Health (Larchmt). 2014;23: 1039-1045.

5. Schenk HM, Jeronimus BF, van der Krieke L, Bos EH, de Jonge P, Rosmalen JGM. Associations of positive affect and negative affect with allostatic load: a lifelines cohort study. Psychosom Med. 2018;80: 160-166.

Comment: Minor comments: Page 11, Line 219: If possible, find an alternate item to use as the example that does not include the racial slurs.

Response: We have replaced this item with an alternate item that does not include racial slurs: “Someone said they expected you to behave certain ways because of your gender and ethnicity (for example, expected you as a Black or Latina woman to wear inappropriate clothes, expected you as an Asian man to be self-controlled and disciplined, as an Asian woman to try to please others, as a Latino man that you would be unfaithful in relationships, etc.)” (line 238 in the version with tracked changes).

Comment: Page 11, Line 224: Describe race/ethnicity categories and rationale for all the participants who were grouped in the other category. Were sensitivity analyses conducted to examine whether including multiracial participants who checked Black into the analyses for participants who solely checked Black? Often multiracial people who are Black and another race are racialized as Black and may have similar stressors.

Response: We thank the reviewer for raising this point. We have added our rationale for including participants who identified with more than one race in a separate category and have also added a sensitivity analysis to examine whether results were influenced by inclusion of multiracial participants who checked Black in the subgroup analysis among Black participants. In the methods section, we state: “Recognizing that Black individuals who identify with at least one other race (i.e., biracial, mixed-race, or multiracial individuals) are often racialized as Black, yet have unique experiences associated with existing at the intersections of multiple racial identities, we conducted these subgroup analyses once among participants who identified exclusively as Black and again among all participants who identified as Black regardless of whether they also identified with another race” (line 331 in the version with tracked changes). In the results section, we report that results were null after inclusion of Black participants who identified with at least one other race: “Associations were attenuated and none remained significant in analyses among the subgroup of all non-Hispanic Black women, which included those who identified exclusively as Black plus those who also identified with another race (data not shown)” (line 494 in the version with tracked changes). In the first paragraph of the discussion, we acknowledge this finding: “our data demonstrate that higher levels of self-reported psychosocial stress were associated with more extreme concentrations of neuroendocrine biomarkers specifically among Black WLWH and HIV-negative women, although this was only the case among monoracial Black participants (and not among all Black participants, including those who identified with at least one other race)” (line 537 in the version with tracked changes).

---

## [Decision Letter · Decision Letter 1]

10 Dec 2021

Psychosocial stress and neuroendocrine biomarker concentrations among women living with or without HIV

PONE-D-20-27314R1

Dear Dr. Levy,

We’re pleased to inform you that your manuscript has been judged scientifically suitable for publication and will be formally accepted for publication once it meets all outstanding technical requirements.

Kind regards,

Yukiko Washio, Ph.D.

Academic Editor

PLOS ONE

Additional Editor Comments (optional):

Reviewers' comments:

Reviewer's Responses to Questions

**Comments to the Author**

1. If the authors have adequately addressed your comments raised in a previous round of review and you feel that this manuscript is now acceptable for publication, you may indicate that here to bypass the “Comments to the Author” section, enter your conflict of interest statement in the “Confidential to Editor” section, and submit your "Accept" recommendation.

Reviewer #1: All comments have been addressed

Reviewer #2: All comments have been addressed

2. Is the manuscript technically sound, and do the data support the conclusions?

Reviewer #1: Yes

Reviewer #2: (No Response)

3. Has the statistical analysis been performed appropriately and rigorously? 

Reviewer #1: Yes

Reviewer #2: (No Response)

4. Have the authors made all data underlying the findings in their manuscript fully available?

Reviewer #1: Yes

Reviewer #2: (No Response)

5. Is the manuscript presented in an intelligible fashion and written in standard English?

Reviewer #1: Yes

Reviewer #2: (No Response)

6. Review Comments to the Author

Reviewer #1: The manuscript has addressed all of my comments and is acceptable for publication in PLOS One. Thank you for allowing me the opportunity to review.

Reviewer #2: (No Response)

7. PLOS authors have the option to publish the peer review history of their article (what does this mean?). If published, this will include your full peer review and any attached files.

Reviewer #1: No

Reviewer #2: No

---

## [Editor Report · Acceptance letter]

14 Dec 2021

PONE-D-20-27314R1 

Psychosocial stress and neuroendocrine biomarker concentrations among women living with or without HIV 

Dear Dr. Levy:

I'm pleased to inform you that your manuscript has been deemed suitable for publication in PLOS ONE. Congratulations! Your manuscript is now with our production department. 

Kind regards, 

on behalf of

Dr. Yukiko Washio 

Academic Editor

PLOS ONE